# Oil Leakage Evaluation for Selection of Emulsion-Based Non-Curable Synthetic Polymer Rubberized Gel (ENC-SPRG) as Waterproofing Material in Underground Structures

**DOI:** 10.3390/ma12233816

**Published:** 2019-11-20

**Authors:** Dong-bum Kim, Su-young Choi, Jin-sang Park, Xing-Yang He, Sang-Keun Oh

**Affiliations:** 1Program of Architecture of Convergence Institute of Biomedical Engineering and Biomaterials of Graduate School, Seoul National University of Science and Technology, 232 Gongneung-ro, Nowon-gu, Seoul 01811, Korea; db2128@naver.com; 2New Material & Convergence Laboratory Co. Ltd., 232 Gongneung-ro, Nowon-gu, Seoul 01811, Korea; csyoung777@gmail.com (S.-y.C.); sciencewater@naver.com (J.-s.P.); 3School of Civil Engineering, Architecture and Environment, Hubei University of Technology, Wuhan 430068, China; hxycn@126.com; 4School of Architecture, Seoul National University of Science & Technology, 232 Gongneung-ro, Nowon-gu, Seoul 01811, Korea

**Keywords:** emulsion-based non-curable synthetic polymer rubberized gel (ENC-SPRG), composite waterproofing material, substrate behavioral movement, oil leakage, filler content settlement, evaluation method

## Abstract

A revised oil leakage evaluation regime is proposed in response to the oil leakage problems of emulsion-based non-curable synthetic polymer rubberized gel (ENC-SPRG) used as a waterproofing material in concrete slabs of residential underground structures. Oil leakage from ENC-SPRG can cause significant economic and environmental damage. As ENC-SPRG waterproofing material is relatively new in the global waterproofing market, a systematic quality control for ENC-SPRG products being manufactured and exported globally is currently non-existent. For the selection of optimal ENC-SPRG, six assessment parameters comprised of averaged and daily average oil leakage mass, averaged and daily average filler content settlement, oil leakage area, and oil leakage duration are proposed. Five ENC-SPRG product specimens are tested to obtain the property values of each parameter. The property values derived from the test results are compared between the tested ENC-SPRG product specimens. With the demonstration of this evaluation regime, a quantified method for a comparative assessment of ENC-SPRG type waterproofing materials is established.

## 1. Introduction

Emulsion-based non-curable synthetic polymer rubberized gel (ENC-SPRG) type waterproofing materials are relatively new in the global waterproofing product market, but are gradually being used for new construction and leakage repair globally in the U.S., Canada, Singapore, China, Korea, etc., with different labels (such as rubberized asphalt mastic, self-adhesive rubberized asphalt polymer, elastomeric asphalt polymer gel, etc.) [1]. In the recently developed ISO TR 16475 Guidelines for the Repair of Water-Leakage Cracks in Concrete Structures, a section on polymerized rubber gel-type waterproofing materials is also present, stressing the advantages of this materials in terms waterproofing performance and joint movement resistance [2]. In China, a new industry standard in the form of JC/T 2428-2017 Non-curable Rubber Modified Asphalt Coating for Waterproofing has recently been developed for standardizing and quality controlling the usage of this type of material for waterproofing in concrete structures.

Emulsion based non-curable synthetic polymer rubberized gel (ENC-SPRG) is a waterproofing material with high resistance to substrate behavioral movement in cracks and joints of concrete slab structures and low-temperature construction conditions [1]. ENC-SPRG is used as a composite waterproofing layer with a sheet membrane. These composite waterproofing layers provide viscoelastic property with high adhesion and high zero-span tensile strength against crack movement caused by thermal change and substrate variation [3]. Due to these properties, ENC-SPRG application is rapidly expanding in the construction field. However, after application, there have been cases where oil components leak out through the cracks and joints of structures, and the necessity of improvement of the stability property of the ENC-SPRG materials is strongly required [3]. The goal of this new evaluation method is to provide the means to secure a safer usage of ENC-SPRG waterproofing that secures long-term durability and oil leakage-free environment in concrete structures.

### 1.1. Understanding of ENC-SPRG Composition as Waterproofing Material

ENC-SPRGs are comprised of binders (asphalt and rubber), additives and fine aggregates (filler contents). ENC-SPRG’s primary advantage comes from the non-hardening property, and that gives the material a non-curable property that allows a high degradation response to substrate behavioral movement of up to 15 mm displacement (crack movement) and low temperature [3]. But in order to ensure that the material can retain this property, emulsification of rubberized polymer and asphalt mastic with aggregate particles is necessary [4]. Refer to Table 1 where the representative common composition ratio of ENC-SPRGs is listed, and Table 2 for the required performance standard of ENC-SPRG waterproofing materials in accordance with the Korea Land and Housing Corporation (Specifications may vary in accordance to different national standards) [5]. Due to these characteristics, ENC-SPRGs are able to respond to concrete joint movement and hydrostatic pressure, and commonly used in concrete slab structures.

### 1.2. Previous Research Works on Leakage Mechanism

#### 1.2.1. Analysis of Filler Content Settlement of ENC-SPRG (1st Stage: Principle Cause of Leakage)

Park et al. [6] research discloses principal causes of filler content settlement in ENC-SPRG relative to the material properties (oil leakage phenomenon). Park et al. present a case study where the leakage mechanism is verified by a filler content (refer to inorganic additives of Table 1) settlement rate measurement. Based on this testing method, the filler content settlement rate was measured and recorded at a daily interval. Park explains that the ENC-SPRG structure forms separate layers within the ENC-SPRG composition, with the resulting supernatant having the highest fluidity with the lowest filler content ratio. Localized zero-tensile stress from crack movement and the compressive load from the concrete substrate and soil layers cause the supernatant (mainly comprised of asphalt component, hereafter called oil) to migrate through the micro-sized pathing and eventually leak through the concrete crack or joint. Based on the results of repeat testing, it was shown that for ENC-SPRG materials tested in this experiment, filler content settlement no longer occurs past the 25th-day interval.

#### 1.2.2. Leakage Pathing Relative to the Crack of Applied Surface (Second Stage: Development of Practical Evaluation)

With only filler content settlement rate measurement data, it is difficult to ascertain which product types have the highest stability performance with regards to oil leakage problems. In light of this background and filler content settlement analysis results of 1.2.1, Park et al. [6] analyzed that another cause of oil leakage is due to the difference in specific gravity between the filler content of the asphalt component caused by demulsification of ENC-SPRG composition in a linear crack of concrete structure. Using another test method outlined in Figure 1, Park et al. confirmed and proved that ENC-SPRG are prone to oil leakage in the presence of a crack in the concrete substrate, and were able to illustrate the oil leakage mechanism in ENC-SPRG composite waterproofing systems as Figure 2.

Through the previous two studies, the cause of oil leakage was identified to be due to material segregation between components of ENC-SPRG itself. However, in the proposed oil leakage evaluation by Park, a specimen with a linear crack is used, and it was not possible to induce a complete oil leakage evaluation from the entire surface of the substrate applied ENC-SPRG [6]. This is apparent as some products that seem to initially pass the previous testing method, resulted in oil leakage in actual construction sites [7]. In the waterproofing construction site, ENC-SRPG is applied to the upper slab of large scale underground concrete structures, there is a large mass of the organic additives generated by the material separation, and the entire surface of the applied ENC-SPRG undergoes the material settlement [8]. This study suggests that for the next stage of the study, a new evaluation method for the comparative decision of optimal ENC-SPRG product is needed.

#### 1.2.3. Requirement for a New Evaluation Method

Due to an absence of a regulation or a standard evaluation criterion to control oil leakage with ENC-SPRG products, information on expected oil leakage mass needs to be accessible in order to assess the potential harm that can be caused to the surrounding environment with a given ENC-SPRG product. Park’s study has only accomplished the analysis and confirmation of the oil leakage mechanism with ENC-SPRG [5]. For a comparative evaluation of ENC-SPRG, a new test method that can conduct a complete oil leakage tendency observation (throughout the entire installed surface) and a corresponding statistical evaluation on the leakage amount, rate and material segregation rate is required. For this study, as an evaluation parameter that can conduct a comparative evaluation of the leakage tendency of the different ENC-SPRG products, six items were proposed: averaged and daily average oil leakage mass and, averaged and daily average filler content settlement, oil leakage area, and oil leakage duration.

## 2. Proposed Oil Leakage Evaluation Method for Comparative Decision of ENC-SPRG

### 2.1. Selection and Explanation of Parameters for Comparative Assessment of ENC-SPRG Specimens

The coming sections outline a proposed leakage evaluation regime and the required parameters that must be taken into consideration for the decision of optimal ENC-SPRG waterproofing systems that have “no leakage” or “the lowest oil leakage mass”. An evaluation regime is proposed to provide the means to comparatively evaluate the oil leakage and filler content settlement tendency (probability and ratio measurement) of different ENC-SPRG product specimens (henceforth specimen) and select the product with the relative optimal stability performance.

#### 2.1.1. Oil Leakage Mass

Oil leakage from the installed ENC-SPRG indicates a higher potential risk of environmental hazard or waterproofing performance. Information on oil leakage mass should be compared during the evaluation to determine and compare each specimen’s leakage potential. Specimens with the least leakage mass indicates higher stability performance with regards to oil leakage tendency. For this parameter, the average mass and the average daily mass need to be recorded during the testing period to understand the potential effects on the environmental hazard, as well as the difficulty in maintenance. First, the cumulative mass of each sample (out of 10) is calculated by adding the daily leakage masses, and the ratio of the total average mass of leaked oil per specimen type is calculated by dividing this value by the installed mass (160 g) of the specimen. Refer to the following Equation (1-a):(1-a)$$\frac{m_{c}}{m_{t}}=m_{r},$$
where:*m_c_*: averaged mass of the sums of leaked oil from the specimen (10 samples) (g),*m_t_*: total mass of ENC-SPRG applied on specimen (160 g, Refer to Table 4)),*m_a_*: averaged mass ratio of leaked oil (%).

For each specimen, the oil leakage duration (number of daily intervals when leakage occurs and the last day in which leakage may occur) can differ. For some specimens, leakage can occur at the beginning of the 25 days testing period, some can occur sporadically. In the case the specimen begins to leak oil, the leakage tendency should be understood by calculating the average daily oil leakage mass. Leakage mass is averaged between the 10 samples for each day when leakage occurs, and the cumulative value of the daily average is divided by the last day during the testing period when oil leakage was observed for each specimen. Refer to the following Equation (1-b):(1-b)$$\frac{m_{d}}{d_{t}}=m_{da},$$
where:*m_d_*: sum of mass of leaked oil averaged at a daily basis among 10 samples per specimens (g),*d_t_*: last day during the testing period when oil leakage was observed (henceforth referred to as ‘leakage range’),*m_da_*: expected average daily oil leakage mass (g).

For comparison between specimens, the specimen with the highest relative daily average oil leakage mass is the basis for the ratio calculation (100%) and the other daily average mass of the other specimens compared to the specimen with the highest daily average mass. Refer to the following Equation (1-c):(1-c)$$\frac{m_{da}}{m_{da\left(specimen\;with\;highest\;mass\right)}}=m_{dr},$$
where:*m_da_*: expected average daily oil leakage mass (g),*m_da_*: (specimen with highest mass): the expected average daily oil leakage mass of the specimen with the highest mass result (%).

#### 2.1.2. Filler Content Settlement Ratio

High filler content settlement indicates a potential loss of waterproofing performance. This is due to that the components in the segregated supernatant lost due to leakage is required to ensure that the ENC-SPRG retains the non-curable property. Aggregate particles settling to the lower section of the ENC-SPRG layer can indicate a high chance of ENC-SPRG layer hardening [9]. Filler content settlement refers to the average filler content different (expected) at the end of the 25 days testing period, and the average degree (percentage) and rate (slope) of filler content settlement need to be recorded to estimate the potential risks on the ENC-SPRG stability performance with regards to material segregation and waterproofing performance. The filler content of the leaked oil is measured as a percentage of the leaked mass for each sample and is then average between the 10 samples. The averaged filler content (%) of the leaked oil and is then divided by original filler content percentage of the specimen. As the pertinent data the degree of deviation from the original filler content of the specimen, the reverse ratio is calculated. Refer to the following Equation (2-a):(2-a)$$1-\frac{Wa}{Wo}=FC_{r},$$
where:W*_a_*: averaged filler content of the leaked oil mass (10 samples) (%),W*_0_*: original filler content of each specimen (refer to Table 3),*FC_r_*: degree of deviation from the original filler content calculated as ratio format (henceforth called filler content settlement ratio) (%).

The expected rate of filler content settlement should also be estimated for each specimen. This is calculated by plotting a linear regression line of the averaged daily filler content percentage over the course of the testing period. At daily intervals when leakage occurred, average filler content percentage is calculated between the 10 samples, and the slope of the regression line is derived. Refer to the following Equation (2-b):(2-b)y=ax+b,
where:*a*: slope,b: intercept,y: filler content percentage,x: leakage day interval

For comparison between specimens, the slope is calculated into a ratio format, and the specimen with the lowest slope is the basis for the ratio calculation (100%) and the other slopes are compared to the specimen with the highest slope. Refer to the following Equation (2-c):(2-c)$$\frac{a}{a_{\left(specimen\;with\;lowest\;slope\right)}}=a_{r},$$
where:*a*: slope*a_r_*: slope calculated in a a ratio format relative to the specimen with the lowest slope.

#### 2.1.3. Oil Leakage Duration Ratio

Filler content settlement induced layer segregation rate does not necessarily indicate how quickly or often oil leakage may occur for each ENC-SPRG. In the case of early concrete cracking after construction, oil leakage may occur earlier or later. Data on oil leakage duration may determine up to approximately how long after installation of ENC-SPRG immediate remedial work or maintenance may have to be performance for this product. This serves to assess how often the ENC-SPRG is expected to leak oil in the first month of installation in the test specimen is formed based on the expected frequency (average number of days when leakage occurs) and duration (last day when oil leakage is observed). The daily intervals when oil leakage is detected is recorded and averaged between the 10 samples. Refer to the following Equation (3):(3)$$\frac{d_{n}}{d_{t}}=d_{r}$$
where:*d_n_*: average number of days when leakage occurred,*d_t_*: last day during the testing period when oil leakage was observed,*d_i_*: oil leakage duration ratio (%).

#### 2.1.4. Oil Leakage Surface Area

Supernatant formation in the installed ENC-SPRG layer may or may not be consistent throughout the entire surface. Defects (cracks, joints, and holes) in the ceiling or wall of underground structures may occur anywhere, and the defects can be a potential outlet for oil leakage [9]. Data on the susceptibility of leakage throughout the ENC-SPRG can be used to determine the likeability of oil leakage occurrence respective to the location of installation and concrete defects. Refer to the following Equation (4):(4)$$\frac{S_{l}}{S_{t}}=S_{r},$$
where:*S_l_*: number of leaked spots present on the specimen,*S_t_*: total number of spots possible to be leaked from the specimen structure (out of 64 in total). Refer to Figure 6b for details.*S_a_*: leakage surface area ratio (%).

## 3. Comparative Evaluation to Obtain the Value of Each Parameter

### 3.1. ENC-SPRG Products for Testing

Five different types of ENC-SPRG products were surveyed in the Korean market. ENC-SPRG specimens that comply with the Korean Industrial Standards (KS F 4917 and KS F 4935) were used. We made and tested 10 samples for each specimen types. Refer to the below Table 3 for details;

### 3.2. Experimental Condition for Testing

#### 3.2.1. Substrate and Waterproofing Condition

Generally, the upper slab of the underground parking lot of residential concrete structures is installed with a protective layer (concrete or mortar) after applying the ENC-SPRG layer [10]. Refer to Figure 3.

For the accurate application of the on-site conditions, the requirement for a weighted load to the test specimen area was calculated using on the basis of the following rational [11]:Waterproofing on concrete THK 100 2.3 t/m^2^,Concrete and top-soil THK 900 1.8 t/m^3^ (top-soil thickness is not constant as THK 800~1000, so the average is 900),The test floor load area is estimated to be 0.07 m^2^,Concrete weight: 0.07 × 0.1 × 2.3 × 1000 = 16.1 kg,Top-soil Weight: 0.07 × 0.9 × 1.8 × 1000 = 113.4 kg,Total: about 130 kg.

#### 3.2.2. Temperature and Load Condition on Waterproofing System

For the temperature condition, the proposed ambient temperature condition during testing is set to 20 °C. In addition, considering the recent summer heat temperature near 40 °C due to the heatwave, it would be appropriate to consider the 40 °C condition to reflect the case where there is little soil depth in the upper part of the underground parking slab [12] (Refer to Figure 3).

For the load condition, the upper slab of the underground parking lot of the multi-floored residential buildings is laminated with protective soil and landscaping on the upper part after waterproofing, which greatly increases the load per unit area [13,14]. In particular, since the waterproof layer is located at the bottom layer, the load is most intensively affected [15].

#### 3.2.3. Testing Period for Observation of Oil Leakage

For the selected five specimens, the oil leakage testing period was set to 25 days. The testing period of 25 days was selected in accordance with Park’s studies where it was confirmed that filler content settlement normally stops after 25 days after installation.

### 3.3. Proposed Test Specimen Applied ENC-SPRG

In order to accommodate the evaluation criteria (parameters) outlined in Section 2, it is important to observe the leakage of different ENC-SPRG for a certain period of time, and also to confirm the differentiation of each specimen. In this study, the new oil leakage testing specimen is developed to overcome the limitation of the leakage characteristic evaluation by the liner crack type specimen (made by Park as shown in Figure 1a above), dimensions of the specimen mold were 270 × 270 × 70 mm. The mortar substrate designated as the bottom layer is drilled at an evenly spaced interval throughout the entire substrate surface that serves as the oil leakage outlet. Using this type of specimen, oil leakage tendency and properties by measuring the oil droplet mass from each hole of the entire ENC-SPRG installed surface for each specimen type can be objectively compared and evaluated. It was judged that the test specimen for oil leakage was suitable for the generation of, and a specially designed acrylic plastic-based specimen structure is used to simulate an environment comprised of an installed ENC-SPRG based composite waterproofing system. The ENC-SPRG material is placed between two mortar substrates. The installation of ENC-SPRG is conducted in accordance with the manufacturer specification. Refer to Figure 4 for illustration.

The fabrication of the test base and the entire specimen assembly procedure is explained and illustrated in Table 4.

### 3.4. Oil Leakage Inducement and Measurement from Specimens

#### 3.4.1. Load Application on Specimen

The susceptibility of five ENC-SPRG specimens’ oil leakage tendencies is evaluated. The test specimen installation is as Table 4, number 4. In the first stage, oil leakage droplet collecting paper is placed under the specimen in the thermal chamber as shown in Figure 5a. A set of weight with the load of 130 kg is placed on top of the specimen, and temperature in the chamber is set at 40 degrees as shown in Figure 5b) (refer to 3.2.2), and the ENC-SPRG flowing out of the hole of the test specimen is observed for 25 days (refer to 3.2.3).

#### 3.4.2. Measurement of Leaked Mass and Area

In the second stage, daily dropped oil mass and position are measured and recorded, as well as the number of spots (below the 64 leakage holes) with the oil leakage drops is counted during 25 days. Refer to Figure 6a for illustrations on the daily oil leakage mass, and Figure 6b for recording of the number and area of daily leakage spots.

#### 3.4.3. Filler Content Measurement

For each of the daily collected oil leaks from the specimens, a filler content measurement testing was conducted. The filler content measurement testing was performed in accordance with the method prescribed in the LH Corporation 42531: 2015 [4]. The leaked oil collected from the collecting paper was placed in a desiccator set to a temperature of 105 ± 5 °C until it reached the constant mass. Afterward, the collected oil was placed in an electric furnace at temperatures ranging from 900–1000 °C. The mass of the aggregate residues (filler content) remaining after maximum heating was calculated once the residue reaches constant mass and cooled in a desiccator (set to 20 ± 3 °C and 65 ± 20% humidity). Refer to Figure 7 below for the illustration of the filler content measurement method.

#### 3.4.4. Assessment of Six Parameters

This is data is taken and the cumulative probability of the over the area of the waterproofing membrane (calculated by the number of the leakage holes of the bottom substrate block of the specimen). ENC-SPRG specimens were evaluated based on six criteria (parameters) in Section 2.1.1; the average oil leakage mass and filler content settlement ratio, the maximum range of leakage over the testing period, area of expected leakage in the installed specimen. The range of daily oil leakage mass and filler content settlement ratio are evaluated. Each criterion derives a ratio as an independent factor and was comprehensively compared between the different specimens.

## 4. Evaluation Results

The results of the respective evaluation criteria based on the relative assessment of five different ENC-SPRG specimens are shown below. Results of the oil leakage mass and filler content settlement measurement of the respective ENC-SPRG specimen by the proposed testing method of Chapter 3 were derived and displayed in the table format below.

### 4.1. Oil Leakage Mass and Filler Content Settlement Measurement

#### 4.1.1. Oil Leakage Mass Measurement Results

Throughout the entire evaluation procedure of 10 samples of the five specimens, the mass of the leaked oil was measured each day where leakage was detected and collected. At the end of the evaluation period (25 days), the collective interval number of days in which leakage occurred for each specimen was recorded. Next, the respective daily mass of the oil leak was added together, and a total average of the leak mass between 10 samples of each specimen was derived. The detailed results of the oil leakage mass and duration for each specimen are as shown below in Table 5, Table 6, Table 7, Table 8 and Table 9 (refer to Note in Table 5, equations apply to all subsequent tables).

Based on the data, the following criteria were evaluated; (1) the averaged oil leakage mass and (2) average daily oil leakage mass for each tested specimen type.

In the case of averaged oil leakage mass, the results were drawn into a bar graph as is shown in Figure 8 below. According to the averaged oil leakage mass over the course of the 25 days testing period, specimen A (24.2 g) and C (40.3 g) had the least leakage mass while specimen D (67.4 g), E (64.1 g) and B (52.6 g) had the highest relative average leakage mass. Specimens with the highest average leakage mass are predicted to have the highest probability of waterproofing performance deterioration through hardening during service life.

The averaged masses of the leaked oil (sum) gained from this new evaluation method provides insight on the tested specimen’s overall expected tendencies regarding performance deterioration due to material loss with regards to potential waterproofing performance that it may apply to the given construction site.

In the case of the average daily leakage mass, the results were drawn into a bar graph as is shown in Figure 9 below. Varying results from the above five data tables disclose that ENC-SPRG oil leakage can occur sporadically throughout the testing period between different specimen types, but two distinct cases were clear (1) cases where leakage occurred at a fixed range of testing days (specimens A, C, D) such that leakage occurs at an earlier period for shorter duration, and (2) cases where leakage occurred for a longer (but discontinuous) period until later into the 25 days of testing period (specimens B, E). For each type of result, different conclusions can be made with regards to maintenance or repair capacity. Specimen D (17.0 g) had the highest average daily oil leakage mass while also having the shortest leakage period, ending the in the earliest (D is four days within the fourth day in Table 8). For these types of tendencies, even though long term leakage problem may not be an issue, immediate remedial action may be required to prevent critical material loss or pollution. Specimens such as C (5.6 g) and E (8.9 g) have relative less average daily leakage mass but occur over a longer period of time (B is 8 days within 22nd day in Table 6 and E is 7 days within 21st day in Table 9). In these cases, immediate remedial action may not be required without the risk of critical loss in waterproofing or material performance, but damage is expected to last longer unless repaired accordingly. Specimens A (3.0 g) and C (5.6 g) can be said to have the most relative stable tendencies in this evaluation, as daily leakage mass is less, and does not last for long with less number of leakage days (A is 8 days within the 10th day in Table 5 and C is six days within the seventh day in Table 7). While performance loss is still expected, tendencies displayed by A and C are most relatively easy to control and perform maintenance to prevent critical performance deterioration. Refer to Figure 9 below for details.

Based on the above results, the ratio calculation for averaged oil leakage mass, and average daily oil leakage mass was derived. Refer to Table 10 below for details.

#### 4.1.2. Filler Content Settlement Results

As a precise measurement of the filler content settlement rate is difficult without destructive testing of the specimens installed with ENC-SPRG samples, a complete assessment of the sedimentation rate cannot be achieved with this evaluation method. For the scope of this testing, the filler content rate of the leaked oil samples was measurement as a reference point of settlement rate by comparing to the original filler content percentage for each specimen type (refer to Table 3 in Section 3.1). The filler content mass of the collected oil leakage samples was measured at a daily interval and the approximate filler content ratio of the oil (leaked supernatant) was recorded. Refer to Table 11, Table 12, Table 13, Table 14 and Table 15 for details (refer to Note in Table 11, equations apply to all subsequent tables).

Based on previous experimental results (refer to Section 1.2.1), it is expected that filler content segregation is expected to occur up to a maximum of 25 days with regards to ENC-SPRG, but a daily filler content measurement of the supernatant in the installed ENC-SPRG could not be conducted without dismantling the specimen. To isolate the information of filler content segregation threshold at which the supernatant oil substance achieves sufficient fluidity to leak through specimen holes, only the filler content ratio measurements from the leaked samples were used to derive a filler content segregation ratio for each specimen types.

Based on the above data, the filler content settlement rate was drawn into a linear digression graph as is shown in Figure 10 below. As can be seen in the graph, the filler contents percentage measured in the leakage droplets is decreasing over the course of the testing period. In the cases of specimen A (−1.06) and C (−1.52), where the leakage mass was relatively the lowest between the specimen types, the slopes of the linear regression lines are still much steeper than specimens B (−0.48) and E (−0.34). In the case of specimen D (−3.08), even though leakage stopped before the 5th day period, filler content is decreasing at a much higher rate than the other specimen types.

This can lead to another conclusion that while specimens with short or early leakage duration may have easier maintenance work if filler content settlement rate is higher (lower slope), it can potentially lead to a higher relative loss of waterproofing performance due to expected hardening of the remaining ENC-SPRG layer in the concrete substrate.

Based on the above results, the ratio calculation for average filler content settlement for each specimen, and filler content settlement rate was derived. Refer to Table 16 below for details;

#### 4.1.3. Statistical Evaluation of Average Oil Leakage Mass and Filler Content Settlement Data

Based on the results outlined from Table 5, Table 6, Table 7, Table 8 and Table 9 (for oil leakage mass) and Table 11, Table 12, Table 13, Table 14 and Table 15 (for filler content settlement), a comprehensive probability density data for the respective 10 specimens over the 25 days of evaluation period were derived, and normative distribution graphs of the respective results were derived to provide a statistical evaluation of the test results. Refer to the below Figure 11a–e for details.

Based on the results of the normative distribution graphs above, the comprehensive frequency of the oil leakage mass (on the left column) and the filler content settlement percentage (on the right column) were derived. In the case of oil leakage mass probability density results, there were varying frequencies, but the maximum leakage loss was never more than 30 g, and the deviation was set to 1 g for mutual comparison purposes. In the results, a wider range in the mass and settlement percentage (*x*-axis) with inconsistent frequency peaks indicate unpredictability of performance. In this regard, it can be said that specimens B, D and E are relatively unpredictable, as well as potential to leak high mass of oil droplets. Specimens A and C have relatively high stability performance in that leakage amounts tend to concentrate at the lower range of mass. Similarly, for the filler content settlement rate, the range and the height of the frequencies determine the stability of the specimen types. While all the specimens show a similar range of settlement rate variance, the settlement rate for all specimens was within 30%, but no less than 10%. Filler content settlement is not necessarily directly correlative to the leakage mass, but generally, a higher percentage of settlement denotes lower stability in terms of long term waterproofing performance. In this regard, specimens with higher frequency at high settlement percentages (B, D, and E) were also the specimens with relative higher unpredictability.

#### 4.1.4. Oil Leakage Duration Ratio Calculation Results

For the oil leakage duration ratio, corresponding data on average number of days when leakage occurred and data on the last day the testing period when oil leakage was observed from Table 5, Table 6, Table 7, Table 8 and Table 9 were taken to derive the oil leakage duration ratio. Refer to Table 17 below for details.

#### 4.1.5. Oil Leakage Surface Area Calculation Results

At the end of each the daily interval when leakage was detected, the average number of points where leakage occurred over the 64 leakage outlet holes was derived and calculated for respective specimens of each specimen types (as in accordance with a grid counting method illustrated in Figure 6b above). With the daily measurements, the oil leakage surface area was calculated as a ratio using Equation (4) from Section 2.1.4. Refer to Table 18 for details.

Based on the above data, it can be inferred that specimens A (66%)and E (41%) will have the least relative probability of oil leakage surface relative to the cracks or defects occurrence in the concrete structure. In contrast, with the case of specimen B (80%) and D (97%), it can be inferred that oil leakage will be localized at cracks or defects occurrence throughout the ENC-SPRG installed surface in waterproofing site.

### 4.2. Comparative Evaluation for Each Specimen Type Oil Leakage Stability Performance

The ratio results calculated using the Equations (1)–(4) were are outlined in Table 19 below, and the respective criteria were comprehensively presented in a radar chart in Figure 12. The variation in the index of each axis indicates the degree of expected properties changes with respect to each degradation factor. A higher percentage indicates a greater degree of likeliness to instability due to exposure to the specific degradation condition {Note: For daily average oil leakage and filler content settlement rate, for comparison purposes, the ratio percentage was calculated relative to the highest values obtained mass the specimens. For both cases, specimen D had the highest daily leakage mass (17.0 g, refer to Figure 8) and filler content settlement rate (slope of −3.08, refer to Figure 7). Other specimens’ results were converted into a percentage ratio based on these highest values.

Based on the above figure, a relative comparison of oil leakage stability performance between different ENC-SPRG specimens can be conducted visually. Specimen A (red line), with the lowest percentage ranges out of the compared specimens, shows the highest stability performance with regards to oil leakage properties. Comparison also points out that while averaged oil leakage mass (mass) ratios may be similarly low when compared to the original mass of the complete ENC-SPRG layer (out of 160 g), the average filler content settlement, leakage period, leakage stability and surface area ratio can differ vastly.

In the case of specimen D (green line) for instance, average leakage mass is high and shows high unpredictability in terms of leakage stability, and also has the highest leakage surface area ratio. However, since the leakage period is concentrated mostly at the beginning of the testing and not lasting no more than four days (as can be seen in Table 3, D), a conclusion can be made that this specimen type’s leakage tendency can be controlled if acted upon in the earlier stages of construction and immediate remedial actions.

In terms of the overall relative stability performance with regards to all six parameters, the following ranking of highest relative stability performance to lowest can be applied for the tested specimens by calculating the respective surface areas of the shapes in the radar chart; A, C, E, B, and D.

## 5. Conclusions

Oil leakage problems of ENC-SPRG materials are still not widespread as this is a relatively new product that is starting to be used globally in the waterproofing market. Among the products currently being used, oil leakage can certainly occur due to low emulsion stability caused by manufacturing or handling issues. Furthermore, ENC-SPRG oil leakage tendencies are generally unpredictable and undocumented, making it crucial to understand the tendencies of the respective products’ evaluation parameters. For the purpose of raising awareness of this issue for future users of this material, an oil leakage evaluation of ENC-SPRG materials is required, and a systematic selection method of optimal ENC-SPRG material to prevent oil leakage problems should be established. The previous method developed by Park et al. discloses the risks of oil leakage when using ENC-SPRG, but the linear cracking and lack of simulation of the environmental condition of an underground structure did not allow for a clear comparative evaluation.

The newly proposed evaluation method accomplishes a comparative evaluation of five ENC-SPRG products based on oil leakage tendencies by assessing the six relevant items that are proposed: averaged and daily average oil leakage mass and, averaged and daily average filler content settlement, oil leakage area, and oil leakage duration. Based on the test results of the 6 parameters, it was made possible to analyze each ENC-SPRG product type’s oil leakage tendency and quantify the expected oil leakage, amount, rate, area, and duration. In this regard, based on the relative comparison, it was clear that specimens A and C had higher stability performance than B, D, and E with regards to oil leakage tendency, and specimen D, in particular, shows the highest instability in terms oil leakage tendency.

Before ENC-SPRG can be considered as an adequate waterproofing material for concrete structures, an evaluation method of optimal ENC-SPRG as is proposed in this paper should be established to provide the means to secure environmental protection and long-term waterproof durability. This new evaluation regime was designed in the hopes that the future development of ENC-SPRG will be accompanied by reminders to avoid the manufacturing or exportation of unregulated ENC-SPRG products with high oil leakage tendency.

## Figures and Tables

**Figure 1 materials-12-03816-f001:**
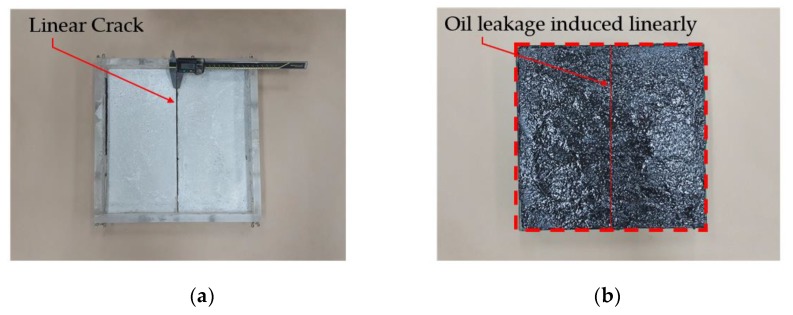
Previous ENC-SPRG evaluation specimen structure; (**a**) linear crack specimen, (**b**) limited observation of oil leakage.

**Figure 2 materials-12-03816-f002:**
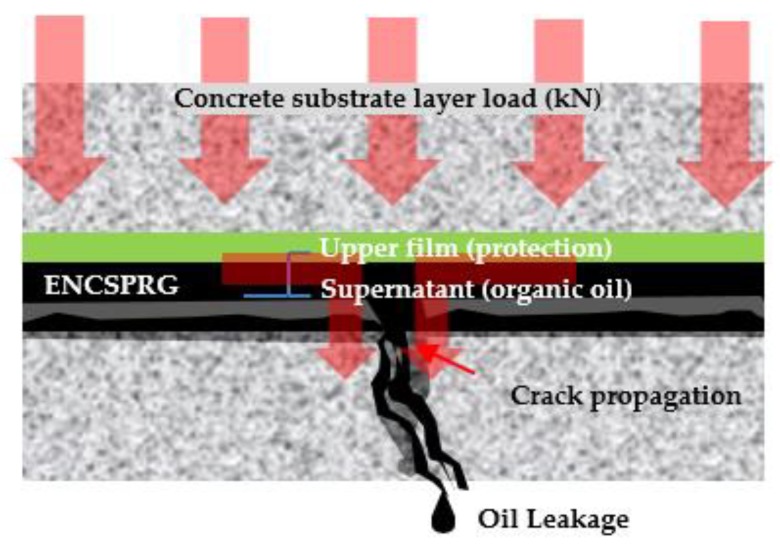
Illustration of oil leakage mechanism of ENC-SPRG in composite waterproofing systems.

**Figure 3 materials-12-03816-f003:**
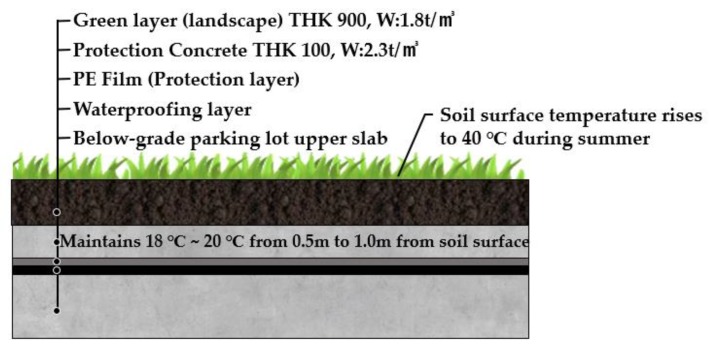
Concrete structure slab layer of ENC-SPRG waterproofing.

**Figure 4 materials-12-03816-f004:**
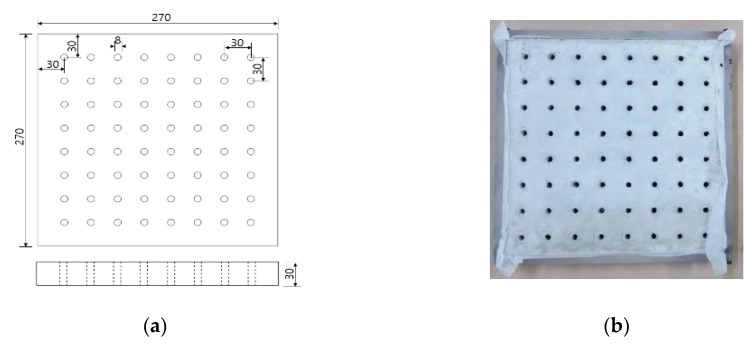
Proposed specimen structure, (**a**) base body dimensions, (**b**) completed specimen base.

**Figure 5 materials-12-03816-f005:**
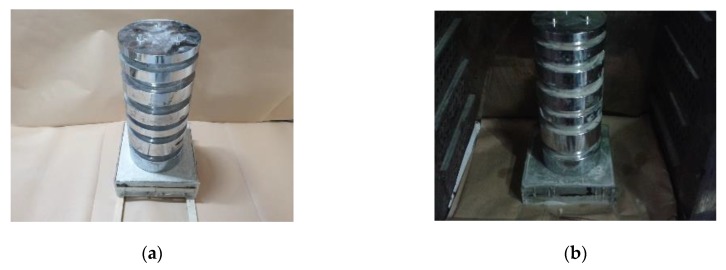
Load conditioning on specimen, (**a**) pressure load applied, (**b**) set in a temperature chamber.

**Figure 6 materials-12-03816-f006:**
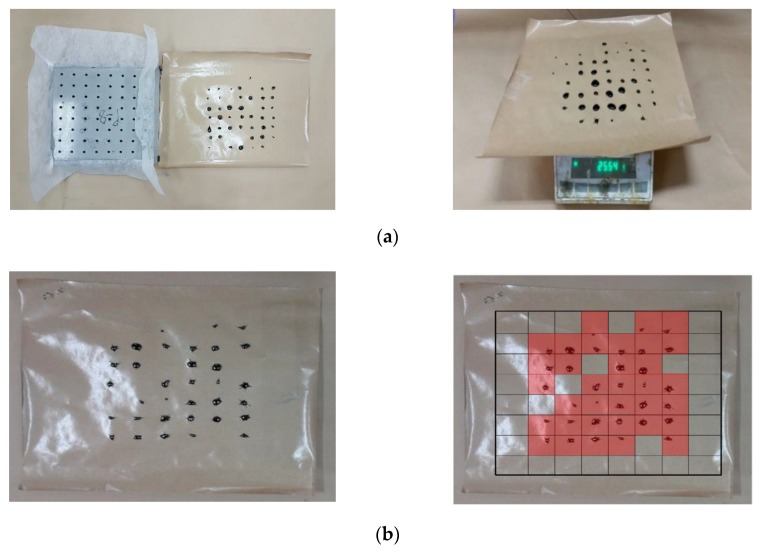
Oil Leakage observation and record; (**a**) oil leak collecting and mass measurement, (**b**) leaked surface area calculation illustrated.

**Figure 7 materials-12-03816-f007:**
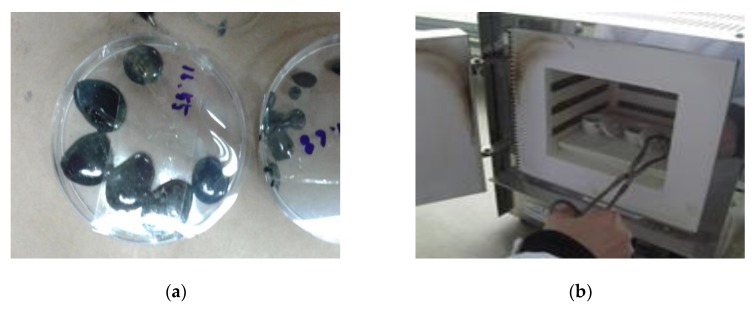
Filler content measurement; (**a**) leaked oil collection, (**b**) set in an electric furnace.

**Figure 8 materials-12-03816-f008:**
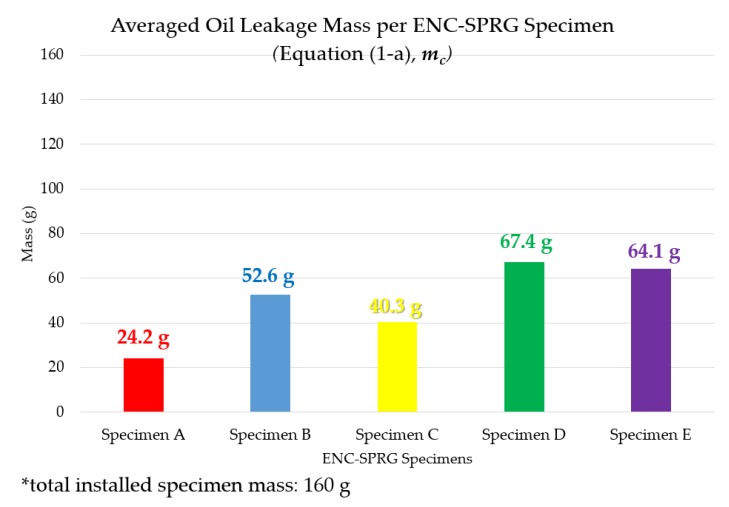
Averaged mass of the sums of leaked oil.

**Figure 9 materials-12-03816-f009:**
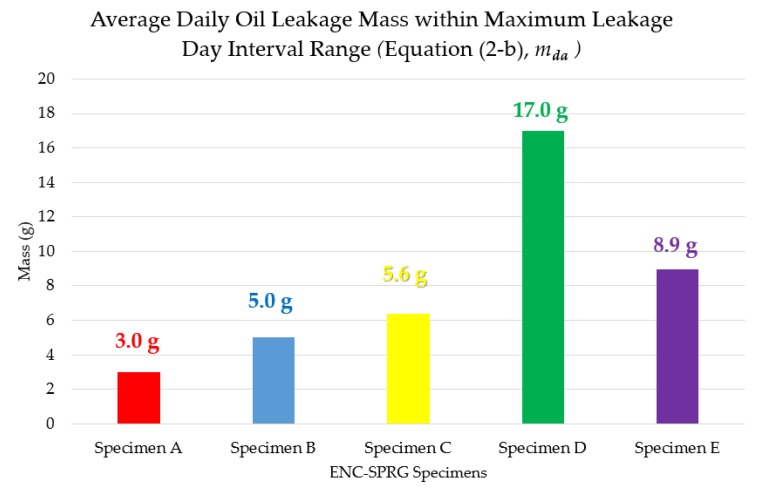
Average daily oil leakage mass of ENC-SPRG specimens.

**Figure 10 materials-12-03816-f010:**
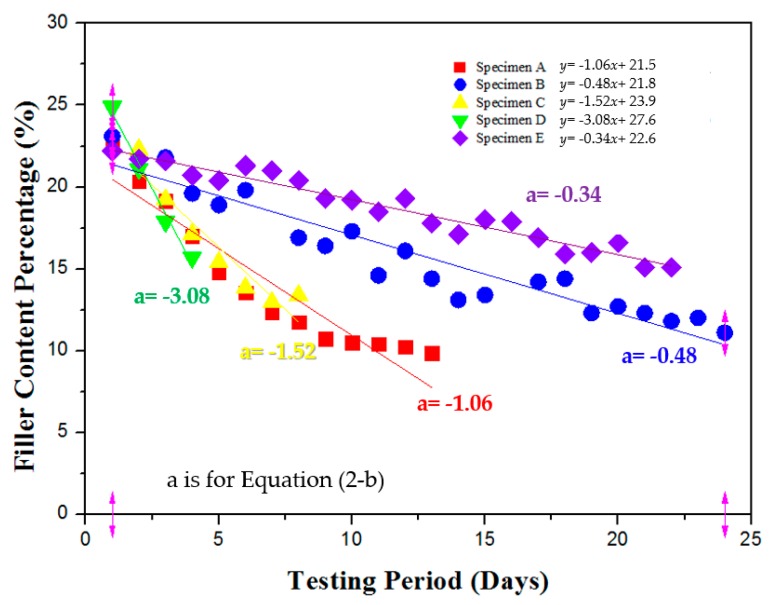
Filler content settlement rate linear regression analysis of ENC-SPRG.

**Figure 11 materials-12-03816-f011:**
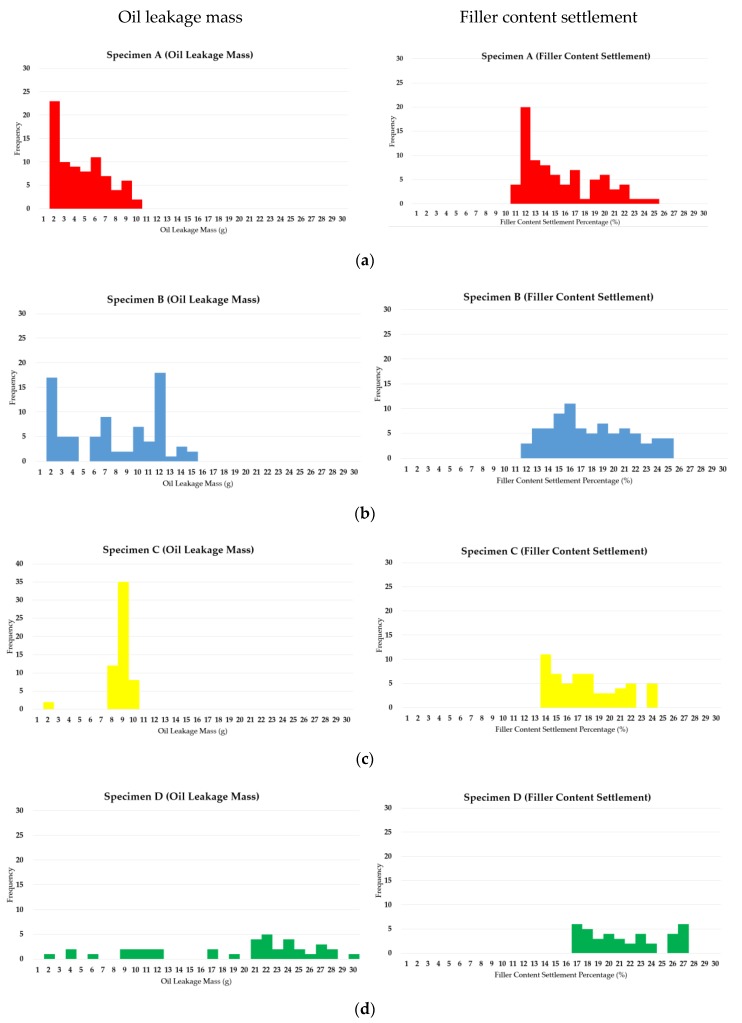
Normative distribution of the probability density calculation of the average oil leakage mass and filler content settlement ratio (left: oil leakage mass, right: filler content settlement); (**a**) specimen A, (**b**) specimen B, (**c**) specimen B, (**d**) specimen D, (**e**) specimen E.

**Figure 12 materials-12-03816-f012:**
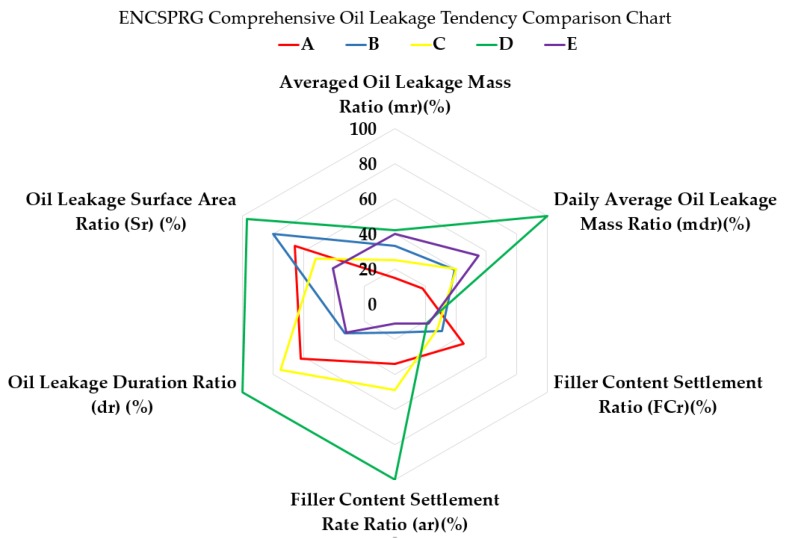
Comprehensive illustration of the evaluation results of the five ENC-SPRG specimens.

**Table 1 materials-12-03816-t001:** General components of emulsion-based non-curable synthetic polymer rubberized gel (ENC-SPRG) (for example).

Components		Specific Gravity	Content Ratio (%)	Remarks
Solids	Asphalt (liquid type)	1.3	40	Organic additives
	Process oil (liquid type)	0.977	8	
	Rubber (solid type)	0.93	15	
	Calcium carbonate (powder type)	2.93	14	Inorganic additives (filler content)
	Natural minerals (powder type)	More than 2	11	
	Other additives	More than 1	2	-
Volatiles	Xylene (liquid type)	Less than 1	10	-
	Toluene (liquid type)			
	Other additives			

**Table 2 materials-12-03816-t002:** LH Construction issued Specification for Performance Standard of ENC-SPRG.

Item	Unit	Quality Standards
1. Solid content	%	>85.0
2. Ash content	%	<15.0
3. Flow Resistance	mm	Flow length <2.0
4. Alkali Resistance	-	No deformation
5. Heat Resistance (60 °C)	-	No deformation
6. Low Temp. Flexibility(−15 °C)	-	No deformation
7. Moisture Content	%	<1.0%
8. Adhesion	N/mm^2^	>0.7
9. Storage Stability	%	Solidity difference within 5%

**Table 3 materials-12-03816-t003:** Five ENC-SPRG specimen properties.

Criteria	Performance Criteria	Product Labels (Specimen)				
		A	B	C	D	E
Solidity (%)	>70.0	78.0	77.8	76.7	72.1	75.3
Filler Content* (%)	>15.0 < 30.0	25.8	24.2	22.5	25.2	23.5
Viscosity (cp)	Brookfield 20 °C Specimen 6 1 rpm, 1 min	340,000	420,000	350,000	346,000	430,333

* The ash mixture ratio (filler content) from this table are used in the coming data analysis in the below sections as *W_0_* in Equation (2-a).

**Table 4 materials-12-03816-t004:** Specimen assembly process.

Step	Illustration	Explanation
1	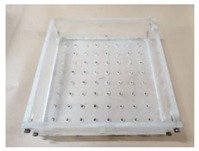	(1) Mold work: Pretreated and fabricated acrylic mold prepared with the leakage outlet holes. For the leakage holes, the allocation of the holes is set to 8 × 8 with 64 holes in total and a 30 mm interval in between the holes. For the convenience of manufacturing, the diameter of the drilling tool was manufactured to Ø9~10 mm, which is included in the tolerance range.
2	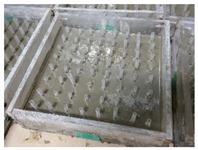	(2) Leakage hole work: Leakage outlet hole molds are prepared by inserting cylindrical tubes (Ø8 × 35 mm) into the acrylic mold holes, and the bottom mortar substrate part is cast. After the substrate layer is cured, the cylindrical tubes are removed.
3	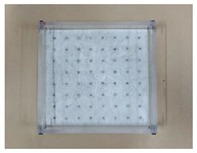	(3) Mortar substrate work: A non-woven fabric of 30 g/m^2^ was fixed on the top of a 270 × 270 mm mortar substrate (T: 30 mm) with Ø8 mm holes spaced at 30 mm intervals (for preventing excess, ENC-SPRG material flow out during ENC-SPRG installation).
4	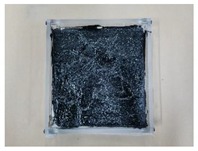	(4) Primer and ENC-SPRG applying: primer is applied evenly at 0.3 kg/m^2^ and cured, and ENC-SPRG application is followed in compliance with the manufacturer specification (either by trowel, brush, metal spatula, or other tools), but the installed mass is fixed to be 160 g (thickness of the ENC-SPRG layer varies, but usually complies to 2 mm).
5	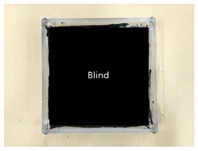	(5) Protection membrane installation; After ENC-SPRG specimen is installed, the protection waterproofing sheet is installed to form a composite waterproofing structure.
6	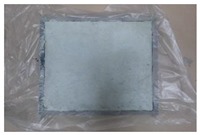	(6) Protection mortar casting; 30 mm thick upper mortar substrate layer (1:3 w/c content) is cast. Specimen curing; specimen is cured for 72 h (for creating enclosed condition).

**Table 5 materials-12-03816-t005:** Oil leakage mass measurement result (specimen A).

	Days	1	2	3	4	5	6	7	8	9	10	11	12	13^1*^	14	15	16	17	18	19	20	21	22	23	24	25	Mass Sum (g)	Avg. (g)	Avg. Days
Samples																													
**A**	**1**	-	6.9	5.2	3.2	1.2	0.2	0.1	0.1	-	-	-	-	-	-	-	-	-	-	-	-	-	-	-	-	-	16.9	24.2^2*^	7
	**2**	-	-	-	7.3	6.2	4.5	3.2	1.2	0.3	0.2	-	-	-	-	-	-	-	-	-	-	-	-	-	-	-	22.9		7
	**3**	-	-	-	-	5.8	4.5	4.0	3.1	1.2	0.9	0.5	0.2	-	-	-	-	-	-	-	-	-	-	-	-	-	20.2		8
	**4**	-	-	-	7.2	5.2	3.2	2.5	2.1	1.2	0.6	0.12	0.1	-	-	-	-	-	-	-	-	-	-	-	-	-	22.2		9
	**5**	8.8	7.6	5.3	4.2	2.8	0.1	-	-	-	-	-	-	-	-	-	-	-	-	-	-	-	-	-	-	-	28.8		6
	**6**	-	7.6	7.1	5.1	4.5	3.9	2.8	2.5	1.52	1.05	0.52	0.2	-	-	-	-	-	-	-	-	-	-	-	-	-	36.8		11
	**7**	-	6.6	5.5	4.1	4.1	2.1	1.5	0.9	-	-	-	-	-	-	-	-	-	-	-	-	-	-	-	-	-	24.8		7
	**8**	-	-	-	-	5.2	4.1	4.1	2.1	1.6	0.6	0.4	0.2	0.1	-	-	-	-	-	-	-	-	-	-	-	-	18.4		9
	**9**	-	-	5.0	4.1	3.1	3.5	2.1	1.4	0.5	-	-	-	-	-	-	-	-	-	-	-	-	-	-	-	-	19.7		7
	**10**	8.1	7.1	6.2	4.6	2.1	1.5	0.8	0.4	0.1	-	-	-	-	-	-	-	-	-	-	-	-	-	-	-	-	30.9		9
	**Daily AVG**	8.5	7.2	5.7	5.0	4.0	2.8	2.3	1.5	0.9	0.7	0.4	0.2	0.1	-	-	-	-	-	-	-	-	-	-	-	-	39.3^3^^*^	3.0^4*^	8^5*^

Note: 1*: *d_t_* for Equations (1-b) and (3), 2*: *m_c_* for Equation (1-a), 3*: *m_d_* used for Equation (1-b), 4*: *m_da_* for Equation (1-b) and (1-c), 5*: *d_n_* for Equation (3).

**Table 6 materials-12-03816-t006:** Oil leakage mass measurement result (specimen B).

	Days	1	2	3	4	5	6	7	8	9	10	11	12	13	14	15	16	17	18	19	20	21	22	23	24^1^^*^	25	Mass Sum (g)	Avg. (g)	Avg. Days
Samples																													
**B**	**1**	-	12.5	-	13.5	5.2	-	-	-	4.3	-	5.2	-	0.5	1.5	2.0	-	-	-	0.5	-	-	-	0.2	-	-	45.4	52.6^2*^	10
	**2**	13.2	12.8	-	-	8.4	-	-	-	10.2	-	10.8	-	-	0.1	-	-	-	-	-	-	0.8	-	-	-	-	56.3		7
	**3**	-	12.9	-	10.4	-	-	-	10.2	-	-	8.5	-	-	-	1.6	-	-	-	-	0.7	-	0.2	-	5.3	-	49.8		8
	**4**	8.2	-	10.3	-	-	10.6	-	-	1.5	-	-	5.2	-	4.2	-	-	4.2	4.1	-	-	-	-	-	-	-	48.3		8
	**5**	-	10.5	-	13.0	-	10.5	-	-	5.6	-	-	5.2	4.2	-	0.3	-	0.2	-	-	-	-	-	-	-	-	49.5		8
	**6**	6.6	10.6	-	-	13.5	10.9	-	2.6	-	-	5.2	-	-	0.6	0.4	-	-	-	-	-	-	-	0.2	-	-	50.6		9
	**7**	9.7	8.5	7.5	-	10.3	5.2	-	-	-	10.5	-	2.3	-	-	-	-	-	0.3	-	-	0.5	-	-	-	-	54.8		9
	**8**	-	10.5	-	10.2	8.9	-	-	-	7.4	-	-	-	8.9	-	-	-	8.6	-	-	-	-	-	0.1	-	-	54.6		7
	**9**	11.1	-	-	-	-	13.2	-	10.5	-	10.5	-	-	-	-	-	-	9.6	-	-	-	2.1	-	-	0.2	-	57.2		7
	**10**	9.5	-	10.6	-	9.9	-	-	-	-	5.6	-	10.4	-	-	2.6	-	6.6	-	2.3	1.2	-	-	0.5	-	-	59.2		10
	**Daily AVG**	9.7	11.2	9.5	11.8	9.4	10.1	0	7.8	5.8	8.9	7.4	5.8	4.5	1.6	1.4	0	5.8	2.2	1.4	1.0	1.1	0.2	0.2	2.7	-	119.5^3*^	5.0^4*^	8^5*^

**Table 7 materials-12-03816-t007:** Oil leakage mass measurement results (specimen C).

	Days	1	2	3	4	5	6	7	8^1*^	9	10	11	12	13	14	15	16	17	18	19	20	21	22	23	24	25	Mass Sum (g)	Avg. (g)	Avg. Days
Samples																													
C	**1**	-	-	7.8	7.3	7.1	7.9	8.0	-	-	-	-	-	-	-	-	-	-	-	-	-	-	-	-	-	-	38.1	40.3^2*^	5
	**2**	-	-	6.9	7.8	7.5	7.2	7.0	-	-	-	-	-	-	-	-	-	-	-	-	-	-	-	-	-	-	36.4		5
	**3**	-	7.5	6.2	8.6	7.6	7.1	7.1	-	-	-	-	-	-	-	-	-	-	-	-	-	-	-	-	-	-	44.1		6
	**4**	-	-	6.1	7.1	7.4	8.3	7.4	-	-	-	-	-	-	-	-	-	-	-	-	-	-	-	-	-	-	36.3		5
	**5**	-	-	7.4	7.2	6.9	7.9	7.1	-	-	-	-	-	-	-	-	-	-	-	-	-	-	-	-	-	-	36.5		5
	**6**	-	8.9	7.2	7.1	7.31	7.8	7.2	0.13	-	-	-	-	-	-	-	-	-	-	-	-	-	-	-	-	-	45.6		7
	**7**	-	-	7.1	7.1	7.04	6.5	8.1	0.10	-	-	-	-	-	-	-	-	-	-	-	-	-	-	-	-	-	35.9		6
	**8**	-	8.1	7.2	7.2	7.03	6.5	7.2	-	-	-	-	-	-	-	-	-	-	-	-	-	-	-	-	-	-	43.2		6
	**9**	-	8.2	7.1	6.5	7.49	7.1	6.5	-	-	-	-	-	-	-	-	-	-	-	-	-	-	-	-	-	-	42.9		6
	**10**	-	8.9	7.0	6.8	7.18	7.2	7.1	-	-	-	-	-	-	-	-	-	-	-	-	-	-	-	-	-	-	44.2		6
	**Daily AVG**	0	8.3	7.0	7.3	7.3	7.4	7.3	0.1	-	-	-	-	-	-	-	-	-	-	-	-	-	-	-	-	-	44.7^3*^	5.6^4*^	6^5*^

**Table 8 materials-12-03816-t008:** Oil leakage mass measurement result (specimen D).

	Days	1	2	3	4^1^^*^	5	6	7	8	9	10	11	12	13	14	15	16	17	18	19	20	21	22	23	24	25	Mass Sum (g)	Avg. (g)	Avg. Days
Samples																													
D	**1**	19.9	21.0	10.6	-	-	-	-	-	-	-	-	-	-	-	-	-	-	-	-	-	-	-	-	-	-	51.5	67.4^2*^	3
	**2**	21.9	22.6	15.3	10.1	-	-	-	-	-	-	-	-	-	-	-	-	-	-	-	-	-	-	-	-	-	69.9		4
	**3**	23.0	22.0	15.1	8.1	-	-	-	-	-	-	-	-	-	-	-	-	-	-	-	-	-	-	-	-	-	68.2		4
	**4**	19.1	20.1	19.5	7.2	-	-	-	-	-	-	-	-	-	-	-	-	-	-	-	-	-	-	-	-	-	65.9		4
	**5**	21.0	22.2	17.6	9.8	-	-	-	-	-	-	-	-	-	-	-	-	-	-	-	-	-	-	-	-	-	70.6		4
	**6**	26.5	26.0	9.9	8.2	-	-	-	-	-	-	-	-	-	-	-	-	-	-	-	-	-	-	-	-	-	70.6		4
	**7**	19.5	25.6	20.5	4.7	-	-	-	-	-	-	-	-	-	-	-	-	-	-	-	-	-	-	-	-	-	70.3		4
	**8**	28.8	25.5	2.2	0.3	-	-	-	-	-	-	-	-	-	-	-	-	-	-	-	-	-	-	-	-	-	56.8		4
	**9**	24.8	23.5	23.5	7.6	-	-	-	-	-	-	-	-	-	-	-	-	-	-	-	-	-	-	-	-	-	79.4		4
	**10**	26.5	20.6	21.5	2.6	-	-	-	-	-	-	-	-	-	-	-	-	-	-	-	-	-	-	-	-	-	71.2		4
	**Daily AVG**	23.1	22.9	15.6	6.5	-	-	-	-	-	-	-	-	-	-	-	-	-	-	-	-	-	-	-	-	-	68.1^3*^	17.0^3*^	4^5*^

**Table 9 materials-12-03816-t009:** Oil leakage mass measurement result (specimen E).

	Days	1	2	3	4	5	6	7	8	9	10	11	12	13	14	15	16	17	18	19	20	21	22^1^^*^	23	24	25	Mass Sum (g)	Avg. (g)	Avg. Days
Samples																													
E	**1**	-	-	-	-	10.6	-	-	-	8.9	7.1	-	10.1	-	11.4	-	-	8.9	-	-	-	-	-	-	-	-	57.0	64.1^2*^	6
	**2**	-	-	-	10.5	-	-	9.8	-	8.5	7.5	-	-	7.5	-	-	-	-	9.5	-	-	-	9.8	-	-	-	63.0		7
	**3**	-	8.8	-	-	-	10.4	-	11.4	-	-	-	9.1	-	-	-	8.1	-	-	9.8	-	-	-	-	-	-	57.6		6
	**4**	-	-	9.5	-	-	11.2	-	10.4	-	-	9.4	-	8.7	-	-	-	-	2.3	-	-	-	10.4	-	-	-	61.9		7
	**5**	-	-	8.2	-	10.0	-	-	10.7	-	-	-	-	8.8	-	-	-	8.3	-	-	-	10.7	-	-	-	-	56.7		6
	**6**	-	9.6	-	-	-	11.5	-	-	10.2	-	-	12.4	-	-	3.5	-	-	6.3	-	-	12.5	-	-	-	-	66.0		7
	**7**	1.5	15.2	-	-	10.5	-	-	-	12.1	-	-	-	9.1	-	-	8.9	-	-	10.8	-	-	5.6	-	-	-	73.7		8
	**8**	-	8.9	10.5	11.5	8.8	-	-	-	-	6.5	-	-	-	-	3.4	-	-	-	-	8.7	-	11.1	-	-	-	69.3		8
	**9**	-	10.2	-	11.5	-	6.5	-	-	8.8	-	-	12.5	-	-	-	10.1	-	-	11.6	-	-	-	-	-	-	71.2		7
	**10**	-	11.4	-	-	10.5	-	-	6.8	-	-	7.8	-	-	11.5	-	-	-	8.2	-	-	-	8.8	-	-	-	64.9		7
	**Daily AVG**	1.5	10.7	9.4	11.1	10.1	9.9	9.8	9.8	9.7	7.0	8.6	11.0	8.5	11.4	3.5	9.0	8.6	6.6	10.7	8.7	11.6	9.1	-	-	-	196.3^3*^	8.9^4*^	7^5*^

**Table 10 materials-12-03816-t010:** Averaged oil leakage mass and average daily oil leakage mass ratio calculation results.

Specimen	Averaged Oil Leakage Mass Ratio		Average Daily Oil Leakage Mass Ratio			
	Averaged Oil Leakage Mass (*m_c_*) (g)	Ratio (Equation (1-a), *m_t_* = 160 g) (*m_r_*) (%)	Mass of Leaked Oil Averaged at a Daily Basis (*m_d_*) (g)	Last Leakage Day (*d_t_* in Equation (3))	Expected Average Daily Oil Leakage Mass (*m_da_*) (g)	Ratio Calculation (Equation (1-c)) (*m_dr_*) (%)
A	24.2	15	39.3	13	3.0	18
B	52.6	33	119.5	24	5.0	29
C	40.3	25	44.7	8	5.6	32
D	67.4	42	68.1	4	17.0 (highest mass)	100
E	64.1	40	196.3	22	8.9	52

**Table 11 materials-12-03816-t011:** Filler content measurement result (specimen A).

	Days	1	2	3	4	5	6	7	8	9	10	11	12	13	14	15	16	17	18	19	20	21	22	23	24	25	Fc%	Avg. (%)
Samples																												
A	**1**	-	20.5	18.9	17.2	15.8	12.6	12.3	12.8	-	-	-	-	-	-	-	-	-	-	-	-	-	-	-	-	-	15.7	14.1^1*^
	**2**	-	-	-	15.3	13.0	11.2	10.5	10.1	10.1	10.0	-	-	-	-	-	-	-	-	-	-	-	-	-	-	-	11.5	
	**3**	-	-	-	-	11.8	11.0	10.6	10.4	10.5	10.3	10.3	10.3	-	-	-	-	-	-	-	-	-	-	-	-	-	10.7	
	**4**	-	-	-	15.2	14.0	13.6	11.0	11.1	11.0	10.8	10.5	10.2	-	-	-	-	-	-	-	-	-	-	-	-	-	11.9	
	**5**	23.1	21.2	19.6	17.2	15.3	14.6	-	-	-	-	-	-	-	-	-	-	-	-	-	-	-	-	-	-	-	18.5	
	**6**	-	19.2	18.8	18.5	16.2	13.5	13.2	12.4	11.7	11.2	10.9	10.5	-	-	-	-	-	-	-	-	-	-	-	-	-	14.2	
	**7**	-	20.2	20.2	18.6	17.6	17.4	15.5	14.2	-	-	-	-	-	-	-	-	-	-	-	-	-	-	-	-	-	17.7	
	**8**	-	-	-	-	13.6	13	12.02	10.5	10.5	10.11	9.9	9.8	9.7	-	-	-	-	-	-	-	-	-	-	-	-	11.0	
	**9**	-	-	18.2	15.9	13.1	13.0	11.6	11.5	10.2	-	-	-	-	-	-	-	-	-	-	-	-	-	-	-	-	13.4	
	**10**	22.6	20.6	19.2	18.2	17.2	15.6	14.5	12.5	11.0	-	-	-	-	-	-	-	-	-	-	-	-	-	-	-	-	16.8	
	**Daily AVG** **^2*^**	22.9	20.3	19.2	17.0	14.8	13.6	12.4	11.7	10.7	10.5	10.4	10.2	9.7	-	-	-	-	-	-	-	-	-	-	-	-	-	

Note: 1*: *W_a_* for Equation (2-a), 2*: data for Equation (2-b).

**Table 12 materials-12-03816-t012:** Filler content measurement result (specimen B).

	Days	1	2	3	4	5	6	7	8	9	10	11	12	13	14	15	16	17	18	19	20	21	22	23	24	25	Fc%	Avg. (%)
Samples																												
B	**1**	-	22.2	-	19.5	18.5	-	-	-	15.2	-	14.3	-	13.0	12.1	11.6	-	-	-	11.2	-	-	-	11.0	-	-	14.9	16.7^1*^
	**2**	23.0	22.2	-	-	17.5	-	-	-	14.8	-	13.3	-	-	11.9	-	-	-	-	-	-	10.9	-	-	-	-	16.2	
	**3**	-	20.7	-	17.9	-	-	-	15.6	-	-	14.3	-	-	-	12.7	-	-	-	-	12.2	-	11.8	-	11.3	-	14.6	
	**4**	23.5	-	21.6	-	-	19.8	-	-	17.2	-	-	15.8	-	15.1	-	-	14.6	14.8	-	-	-	-	-	-	-	17.8	
	**5**	-	23.8	-	22.3	-	20.9	-	-	18.5	-	-	15.4	14.8	-	14.4	-	13.7	-	-	-	-	-	-	-	-	18.0	
	**6**	21.9	21.6	-	-	20.0	18.6	-	17.6	-	-	16.5	-	-	13.1	13.5	-	-	-	-	-	-	-	10.9	-	-	17.1	
	**7**	23.6	23.1	22.9	-	20.2	19.9	-	-	-	18.2	-	16.8	-	-	-	-	-	14.1	-	-	13.3	-	-	-	-	19.1	
	**8**	-	20.8	-	18.6	17.8	-	-	-	16.3	-	-	-	15.5	-	-	-	15.0	-	-	-	-	-	13.7	-	-	16.8	
	**9**	23.1	-	-	-	-	19.8	-	17.3	-	16.6	-	-	-	-	-	-	13.5	-	-	-	12.7	-	-	11.0	-	16.3	
	**10**	23.7	-	20.8	-	19.3	-	-	-	-	17.1	-	16.4	-	-	14.7	-	14.1	-	13.4	13.3	-	-	12.3	-	-	16.5	
	**Daily AVG^2*^**	23.1	22.1	21.8	19.6	18.9	19.8	-	16.8	16.4	17.3	14.6	16.1	14.4	13.1	13.4	-	14.2	14.5	12.3	12.8	12.3	11.8	12.0	11.2	-	-	

**Table 13 materials-12-03816-t013:** Filler content measurement result (specimen C).

	Days	1	2	3	4	5	6	7	8	9	10	11	12	13	14	15	16	17	18	19	20	21	22	23	24	25	Fc%	Avg. (%)
Samples																												
C	**1**	-	-	20.8	16.3	15.5	13.4	12.9	-	-	-	-	-	-	-	-	-	-	-	-	-	-	-	-	-	-	15.8	16.2^1*^
	**2**	-	-	16.2	16.2	14.8	12.2	12.2	-	-	-	-	-	-	-	-	-	-	-	-	-	-	-	-	-	-	14.3	
	**3**	-	22.4	18.8	15.3	13.2	12.7	12.1	-	-	-	-	-	-	-	-	-	-	-	-	-	-	-	-	-	-	15.8	
	**4**	-	-	20.8	19.1	16.3	16.1	14.2	-	-	-	-	-	-	-	-	-	-	-	-	-	-	-	-	-	-	17.3	
	**5**	-	-	19.9	17.8	15.7	14.8	12.3	-	-	-	-	-	-	-	-	-	-	-	-	-	-	-	-	-	-	16.1	
	**6**	-	22.2	18.4	18.3	14.5	13.1	13.0	12.8	-	-	-	-	-	-	-	-	-	-	-	-	-	-	-	-	-	16.0	
	**7**	-	-	20.2	20.0	19.1	16.6	15.6	14.0	-	-	-	-	-	-	-	-	-	-	-	-	-	-	-	-	-	17.6	
	**8**	-	22.3	19.7	15.5	13.2	12.8	12.1	-	-	-	-	-	-	-	-	-	-	-	-	-	-	-	-	-	-	15.9	
	**9**	-	22.5	20.3	17.2	16.8	13.3	12.1	-	-	-	-	-	-	-	-	-	-	-	-	-	-	-	-	-	-	17.0	
	**10**	-	22.1	17.2	15.4	15.1	13.4	13.2	-	-	-	-	-	-	-	-	-	-	-	-	-	-	-	-	-	-	16.1	
	**Daily AVG^2*^**	-	22.3	19.2	17.1	15.4	13.8	13.0	13.4	-	-	-	-	-	-	-	-	-	-	-	-	-	-	-	-	-	-	

**Table 14 materials-12-03816-t014:** Filler content measurement result (specimen D).

	Days	1	2	3	4	5	6	7	8	9	10	11	12	13	14	15	16	17	18	19	20	21	22	23	24	25	Fc%	Avg. (%)
Samples																												
D	**1**	25.1	19.3	16.1	-	-	-	-	-	-	-	-	-	-	-	-	-	-	-	-	-	-	-	-	-	-	20.2	20.0^1*^
	**2**	24.86	21.4	17.6	15.9	-	-	-	-	-	-	-	-	-	-	-	-	-	-	-	-	-	-	-	-	-	19.9	
	**3**	25.13	22.5	18.3	15.6	-	-	-	-	-	-	-	-	-	-	-	-	-	-	-	-	-	-	-	-	-	20.4	
	**4**	24.71	21.9	17.7	15.3	-	-	-	-	-	-	-	-	-	-	-	-	-	-	-	-	-	-	-	-	-	19.9	
	**5**	25.04	21.5	18.1	16.1	-	-	-	-	-	-	-	-	-	-	-	-	-	-	-	-	-	-	-	-	-	20.2	
	**6**	25.18	20.1	17.6	16.1	-	-	-	-	-	-	-	-	-	-	-	-	-	-	-	-	-	-	-	-	-	19.7	
	**7**	24.13	19.9	16.5	15.1	-	-	-	-	-	-	-	-	-	-	-	-	-	-	-	-	-	-	-	-	-	18.9	
	**8**	25.11	22.1	19.6	15.5	-	-	-	-	-	-	-	-	-	-	-	-	-	-	-	-	-	-	-	-	-	20.6	
	**9**	24.92	20.7	18.5	16.7	-	-	-	-	-	-	-	-	-	-	-	-	-	-	-	-	-	-	-	-	-	20.2	
	**10**	25.18	21.1	18.4	15.1	-	-	-	-	-	-	-	-	-	-	-	-	-	-	-	-	-	-	-	-	-	19.9	
	**Daily AVG^2*^**	24.9	21.1	17.8	15.7	-	-	-	-	-	-	-	-	-	-	-	-	-	-	-	-	-	-	-	-	-	-	

**Table 15 materials-12-03816-t015:** Filler content measurement result (specimen E).

	Days	1	2	3	4	5	6	7	8	9	10	11	12	13	14	15	16	17	18	19	20	21	22	23	24	25	Fc%	Avg. (%)
Samples																												
E	**1**	-	-	-	-	22.4	-	-	-	21.2	20.9	-	19.7	-	18.1	-	-	16.3	-	-	-	-	-	-	-	-	19.8	18.9^1*^
	**2**	-	-	-	21.1	-	-	21.0	-	17.6	17.3	-	-	15.9	-	-	-	-	14.2	-	-	-	14.4	-	-	-	17.4	
	**3**	-	22.2	-	-	-	22.4	-	21.6	-	-	-	20.6	-	-	-	19.1	-	-	16.2	-	-	-	-	-	-	20.4	
	**4**	-	-	22.1	-	-	21.9	-	21.1	-	-	19.2	-	18.7	-	-	-	-	17.1	-	-	-	15.4	-	-	-	19.4	
	**5**	-	-	21.4	-	21.1	-	-	20.5	-	-	-	-	19.6	-	-	-	17.5	-	-	-	15.2	-	-	-	-	19.2	
	**6**	-	22.9	-	-	-	20.2	-	-	19.4	-	-	18.4	-	-	17.6	-	-	16.9	-	-	15.1	-	-	-	-	18.6	
	**7**	22.2	20.4	-	-	19.5	-	-	-	18.9	-	-	-	17.1	-	-	16.4	-	-	15.6	-	-	15.0	-	-	-	18.1	
	**8**	-	21.6	21.3	20.2	19.4	-	-	-	-	19.5	-	-	-	-	18.3	-	-	-	-	16.6	-	15.4	-	-	-	19.0	
	**9**	-	22.8	-	20.7	-	20.9	-	-	19.2	-	-	18.4	-	-	-	18.1	-	-	16.1	-	-	-	-	-	-	19.5	
	**10**	-	20.3	-	-	19.6	-	-	18.5	-	-	17.9	-	-	16.1	-	-	-	15.4	-	-	-	15.1	-	-	-	17.6	
	**Daily AVG^2*^**	22.2	21.7	21.6	20.7	20.4	21.4	21.0	20.4	19.3	19.2	18.6	19.3	17.8	17.1	18.0	17.9	16.9	15.9	16.0	16.6	15.2	15.1	-	-	-	-	

**Table 16 materials-12-03816-t016:** Averaged oil leakage mass and average daily oil leakage mass ratio calculation results.

Specimen	Filler Content Settlement Ratio		Filler Content Settlement Rate Ratio	
	Averaged Filler Content of the Leaked Oil Mass (*w_a_*) (%)	Ratio (Equation (2-a), for *w_0_* Refer to Table 3) (*FC_r_*) (%)	Slope from Linear Regression Analysis (*a*)	Ratio (Equation (2-c)) (*a_r_*) (%)
A	14.1	45	−1.06	34
B	16.7	31	−0.48	16
C	16.2	28	−1.52	49
D	20.0	21	−3.08 (lowest slope)	100
E	18.9	20	−0.34	11

**Table 17 materials-12-03816-t017:** Oil leakage duration ratio calculation results.

Specimen	Number of Leakage Days (Avg.) (*d_n_* in Equation (3))	Last Leakage Day (*d_t_* in Equation (3))	Oil Leakage Duration Ratio (Equation (3)) (*d_r_*) (%)
A	8	13	62
B	8	24	33
C	6	8	75
D	4	4	100
E	7	22	32

**Table 18 materials-12-03816-t018:** Average oil leakage surface area calculation results.

Specimen	Number of Leakage Points (Avg.) (*S_l_*)	Oil Leakage Surface Area Ratio (Equation (4)) (*S_l_*/64 = *S_r_*) (%)
A	42	66
B	51	80
C	33	52
D	62	97
E	26	41

**Table 19 materials-12-03816-t019:** Ratio results of each evaluation criterion.

Equation	(1-a)	(1-c)	(2-a)	(2-b)	(3)	(4)
Specimen	Averaged Oil Leakage Mass Ratio, *m_r_* (%)	Daily Average Oil Leakage Mass Ratio, *m_dr_* (%)	Filler Content Settlement Ratio, *FC_r_* (%)	Filler Content Settlement Rate Ratio, *Slope a_r_* (%)	Maximum Leakage Duration Range Ratio, *d_r_* (%)	Oil Leakage Surface Area Ratio, *S_r_* (%)
A	15	18	45	34	62	66
B	33	29	31	16	33	80
C	25	32	28	49	75	52
D	42	100	21	100	100	97
E	40	52	20	11	32	41

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
