# Peer review of "Oil Leakage Evaluation for Selection of Emulsion-Based Non-Curable Synthetic Polymer Rubberized Gel (ENC-SPRG) as Waterproofing Material in Underground Structures"

_materials, 2019, doi:10.3390/ma12233816_

Round 1
Reviewer 1 Report
This is a study of sealing products from Korea. It is not clear to me how widespread is the use of these products outside of that country - if it isnt then it will very low reader interest and should be rejected. That must be clarified by the authors.
It looks at how to use experiments evaluate oil leakage from these products. This is worth while and seems to be handled OK. However the evaluation feels poorly done . they do multiple tests on each but no statistical evaluation, and quote averages to a silly degree of precision e.g. 14.20 when the numbers used were between 14 and 22.
To be published should show some relevance to workers outside Korea and a proper statistical evaluation with results showing uncertanties.
Author Response
Thank you kindly for your reviews. Your comments have been applied accordingly to improve the quality and content of our paper. We have provided the following response to your comments to the best of our understanding;
Reviewer comment 1: This is a study of sealing products from Korea. It is not clear to me how widespread is the use of these products outside of that country - if it isn’t then it will very low reader interest and should be rejected. That must be clarified by the authors.
A:
There are certainly ENC-SPRG products being manufactured and used outside of Korea, and it was not our intent to limit the scope of this paper’s field to Korea only. There are cases of usage of this materials in Singapore, China, U.S., and Canada (with different names of the material type and a terminology for this type of materials has not yet been standardized), all of which have had minor cases of oil leakage, but such details have not been officially publicized due to legal restrictions. Furthermore, as of recent, new standards are also being made and published (such as ISO TR 16475 and JC/T 2428 in China) for quality control of these type of materials (this point was added to the paper). However, in the current state of affairs, ENC-SPRG is a relatively new type of waterproofing materials, and this is especially the case when used in underground structures. There are some national standards and specifications on this material, but a standardized testing method that specialize in oil leakage tendency property does not currently exist for quality control and disclosing this problem to the general public. Websites on overseas products currently located in the US are provided below.
https://gcpat.com/en
http://en.re-new.co.kr/products/turbo-seal/
Theoretically, the material is considered to be an excellent alternative to existing other products (PVC sheets, urethane, cementitious grouts, etc.), but in reality, it is difficult to ensure that ENC-SPRG will be able to uphold the same level of performance or quality in actual sites as is shown in laboratory settings. Early demulsification or material degradation may occur during storage, preparation, and construction due to lack of awareness of the material property. Some ENC-SPRG products take mishandling of the product in consideration during manufacturing process, but newly developing manufacturers are simply unaware or disregard these issues entirely, but this situation is not yet documented in the media or in the academic field. When oil leakage is detected, they are usually repaired, but these cases are rarely documented or made public for statistical research purposes. In some cases, research centers such as ours in Seoul National University of Science and Technology come across such data for material analysis and testing reports when requested by the companies, but we are unable to open these data to the public due to copyright regulations, which is why only data concerning oil leakage in Korea were used in the previous version of this paper. In Korea, there are statistical data available for such problems because of the recent trend of regulating and strengthening environmental protection in residential areas and buildings, and companies manufacturing and patenting new products (such as ENC-SPRG) are being strictly enforced to open all cases of product usage, which is why were able to use the statistical data in the first place.
Regardless, parts of the text concerning the application range of this topic (limitation to Korea) has been removed to avoid confusion about the applicability of this study, and sections have been added to elaborate that ENC-SPRG material concerns a global market. Please refer to the first part of Section 1. Introduction in the revised paper for more details.
Reviewer comment 2: It looks at how to use experiments evaluate oil leakage from these products. This is worth while and seems to be handled OK. However the evaluation feels poorly done. they do multiple tests on each but no statistical evaluation, and quote averages to a silly degree of precision e.g. 14.20 when the numbers used were between 14 and 22.
A: The decimal points have been moved up by one digit to remove unnecessarily precise numbering of the data. A separate section to provide a statistical evaluation (now titles Section 4.1.3 Statistical evaluation of average oil leakage mass and filler content settlement data) has been included in the paper. Furthermore, the ordering in which the data is presented has been changed throughout the paper such that the quality of the evaluation has been improved and presented more clearly. Please refer to the revised paper for more details.
Please be advised that the contents of the paper concerning points outside of what has been addressed by the reviewer has been revised to accord with the revision points made by other reviewers as well. Please refer to the revised paper for details.
Reviewer 2 Report
The authors reported the oil leakage evaluation in residential underground structures caused by usage of emulsion based non-curable synthetic polymer rubberized gel (ENC-SPRG) as a part of composite waterproofing layer, using five ENC-SPRG products used in Korea market. The paper is divided into 5 sections and I have some remarks:
Section 1:
-It is very long and basically it is dedicated to describing the previous work reported by Park et al 2018.
-The work is focused on statistics of South Korea, but what about the rest of the work?
-In 1.1, in the first sentences say that ENC_SPRG is a waterproofing material product markert but is global or in South Korea?
-In 1.2.3, is where the object of the work is described, this subsection needs to improve.
Section 2:
-2.1, This section is introductory.
-2.1.1 to 2.1.4 y section 2.2 can be mixed to improve the explanation
-2.2.1 the equations (1-a,1-b) described there does not represent the average but the ratio.
-Equation 2-b is the linear fit that is well known does not need much explanation
-2.2.4, Equation 4 is not correct
Section 4,
The results showed are obtained using an incorrect formula or the equations described in section 2 are not well explained
-Error not reported.
-Figures 11 and 12 can be changed for tables and improved, like including the total mass which is only shown on page 7. Also, do not confuse the terms "products" with "specimen", check table 15, 16.
Section 5,
Must be improved
Author Response
Thank you kindly for your reviews. Your comments have been applied accordingly to improve the quality and content of our paper. We have provided the following response to your comments to the best of our understanding;
Reviewer comment 1: The authors reported the oil leakage evaluation in residential underground structures caused by usage of emulsion based non-curable synthetic polymer rubberized gel (ENC-SPRG) as a part of composite waterproofing layer, using five ENC-SPRG products used in Korea market. The paper is divided into 5 sections and I have some remarks:
Section 1:
Reviewer comment : It is very long and basically it is dedicated to describing the previous work reported by Park et al 2018.
A: Introduction has been shortened, and previous work reported by Park et al has been abbreviated significantly
Reviewer comment: The work is focused on statistics of South Korea, but what about the rest of the work? In 1.1, in the first sentences say that ENC_SPRG is a waterproofing material product market but is global or in South Korea?
A:
There are certainly ENC-SPRG products being manufactured and used outside of Korea, and it was not our intent to limit the scope of this paper’s field to Korea only. There are cases of usage of this materials in Singapore, China, U.S., and Canada (with different names of the material type and a terminology for this type of materials has not yet been standardized), all of which have had minor cases of oil leakage, but such details have not been officially publicized due to legal restrictions. Furthermore, as of recent, new standards are also being made and published (such as ISO TR 16475 and JC/T 2428 in China) for quality control of these type of materials (this point was added to the paper). However, in the current state of affairs, ENC-SPRG is a relatively new type of waterproofing materials, and this is especially the case when used in underground structures. There are some national standards and specifications on this material, but a standardized testing method that specialize in oil leakage tendency property does not currently exist for quality control and disclosing this problem to the general public. Websites on overseas products currently located in the US are provided below.
https://gcpat.com/en
http://en.re-new.co.kr/products/turbo-seal/
Theoretically, the material is considered to be an excellent alternative to existing other products (PVC sheets, urethane, cementitious grouts, etc.), but in reality, it is difficult to ensure that ENC-SPRG will be able to uphold the same level of performance or quality in actual sites as is shown in laboratory settings. Early demulsification or material degradation may occur during storage, preparation, and construction due to lack of awareness of the material property. Some ENC-SPRG products take mishandling of the product in consideration during manufacturing process, but newly developing manufacturers are simply unaware or disregard these issues entirely, but this situation is not yet documented in the media or in the academic field. When oil leakage is detected, they are usually repaired, but these cases are rarely documented or made public for statistical research purposes. In some cases, research centers such as ours in Seoul National University of Science and Technology come across such data for material analysis and testing reports when requested by the companies, but we are unable to open these data to the public due to copyright regulations, which is why only data concerning oil leakage in Korea were used in the previous version of this paper. In Korea, there are statistical data available for such problems because of the recent trend of regulating and strengthening environmental protection in residential areas and buildings, and companies manufacturing and patenting new products (such as ENC-SPRG) are being strictly enforced to open all cases of product usage, which is why were able to use the statistical data in the first place.
Regardless, parts of the text concerning the application range of this topic (limitation to Korea) has been removed to avoid confusion about the applicability of this study, and sections have been added to elaborate that ENC-SPRG material concerns a global market. Please refer to the first part of Section 1. Introduction in the revised paper for more details.
Reviewer Comment: In 1.2.3, is where the object of the work is described, this subsection needs to improve.
A: 1.2.3 has been revised and improved to be more clear. Evaluation items have been outlined more clearly, and purpose of the evaluation was made more clear.
Section 2:
Reviewer Comment: 2.1, This section is introductory.
A: 2.1 has been moved to the introduction section and revised to be shortened
Reviewer Comment: 2.1.1 to 2.1.4 y section 2.2 can be mixed to improve the explanation
A: Sections 2.1.1 to 2.1.4 has been combined with Section 2.2 and the corresponding subsections to improve the explanation (section 2.2 has been removed)
Reviewer Comment: 2.2.1 the equations (1-a,1-b) described there does not represent the average but the ratio.
A: Equations (1-a and 1-b) have been amended such that they now explicate about the ratio rather than the average.
Reviewer Comment: Equation 2-b is the linear fit that is well known does not need much explanation
A: Unnecessary explanation has been removed, but the equation has been left in the paper for clarification purposes as the slope of the linear regression is required for comparing the rate filler content settlement between the different ENC-SPRG specimens.
Reviewer Comment: 2.2.4, Equation 4 is not correct
A: Explained the terms in Equation 4 with more clarity.
Section 4,
Reviewer Comment: The results showed are obtained using an incorrect formula or the equations described in section 2 are not well explained
A: formula have been revised such that they now display the variable labels correctly. Notes in Tables 5-9 and 10-14 have been revised as well. Please refer to the new Section 2.1 (Subsections 2.1.1 to 2.1.4) for details on the revised formula variables.
Reviewer Comment: Error not reported.
A: error margins have been corrected accordingly to the above
Reviewer Comment: Figures 11 and 12 can be changed for tables and improved, like including the total mass which is only shown on page 7. Also, do not confuse the terms "products" with "specimen", check table 15, 16.
A: The terms have been revised (products have been changed to specimen), and Figures 11 and 12 (now figures 7 and 8 respectively in page 16 and 17) have been amended to provide more specific details about the total installed specimen mass (160 g). Refer to Figure 7, and 8 for details. Texts in the paper have been adjusted accordingly as well. When concerning the materials used for the testing (materials with labels A, B, C, D, and E) are called specimen (shortened from Product Specimen), and for cases when addressing the materials in a general term (being used in construction site), materials are now called products (without A, B, C, D, or E labels)
Section 5
Reviewer Comment: Must be improved
A: Conclusion (Section 5) has been revised and made more clearly to better suit the results of the study.
Please be advised that the contents of the paper concerning points outside of what has been addressed by the reviewer has been revised to accord with the revision points made by other reviewers as well. Please refer to the revised paper for details.
Reviewer 3 Report
Very specialized article.
It is better to formulate conclusions. Indicate the advantages and new elements of the oil leakage assessment method.
Shorten article title.
Author Response
REVIEWER 3
Thank you kindly for your reviews. Your comments have been applied accordingly to improve the quality and content of our paper. We have provided the following response to your comments to the best of our understanding;
It is better to formulate conclusions. Indicate the advantages and new elements of the oil leakage assessment method.
A: Conclusion has been revised and made more clearly to better suit the results of the study.
Shorten article title.
A: Article title has been shortened, now provisionally titled: Oil Leakage Evaluation for Selection of Emulsion based Non-curable Synthetic Polymer Rubberized Gel (ENC-SPRG) as waterproofing materials in Underground Structures
Please be advised that the contents of the paper concerning points outside of what has been addressed by the reviewer has been revised to accord with the revision points made by other reviewers as well. Please refer to the revised paper for details.
Round 2
Reviewer 1 Report
The authors have clairified the use of teh products outside of Korea, and have improved the statistical component. As it is a potentially important product I am happy for it to be published
Author Response
Thank you kindly for your comments, review, and contribution to this paper. We are very grateful.
Reviewer 2 Report
Dear Authors,
Thank you very much for considering my previous comments. The work has been improved. Still, I have some remarks:
The paper is divided into 5 sections.
Section 1:
-Was improved.
Section 2:
-2.1.1 to 2.1.4 need to improve.
-some equations still not correct please check it.
Section 4,
The results showed are obtained using an incorrect formula or the equations described in section 2 are not well explained
-Error not reported.
Section 5,
Must be improved
Author Response
Thank you kindly once more for your continued efforts and contribution in the making of this article. We are very grateful for your feed backs.
We have made further revisions to the paper in accordance to your previous comments, we hope we can trouble you to take a look at the paper once more.
We attached the following individual response to your comments;
The paper is divided into 5 sections.
Section 1:
Reviewer comment:
-Was improved.
Response:
Thank you kindly for your comment.
Section 2:
Reviewer Comment:
-2.1.1 to 2.1.4 need to improve.
-some equations still not correct please check it.
Response:
We made significant changes to sections 2.1.1 and 2.1.4. We profusely apologize for the equation errors, especially with the variable labels. We checked several times this time around so we are certain that there should no longer be incorrect labels to the equation. We are sorry for making such basic mistakes.
Section 4,
Reviewer Comment:
The results showed are obtained using an incorrect formula or the equations described in section 2 are not well explained
-Error not reported.
Response:
In the process of revising the Equations in Section 2, we did in fact notice that the results shown did have some incorrect calculations. These have been amended accordingly. Also, to prevent any chances of confusion, we have added new tables at the end of each subsections on parameter calculations showing the process of obtaining the ratios, which will all be compiled at the last comparative evaluation in (newly numbered) Table 19. If the previous versions were poorly written such that it was difficult to understand the calculation process, we apologize once more, and hope that this revision will have made the flow of the paper more clear to understand. Data figures and graphs have also been amended accordingly (Figures 8, 9 and 11). However, we are not entirely sure what you mean by "error not reported." If this means that there is error with the calculations, then we hope that with the revisions, there will no long be any error with the results. Please refer to the revised (and newly added) Tables 5, 10, 16, 17, 18, and 19 in the revised paper for details.
Reviewer comment:
Section 5,
Must be improved
Response:
Section 5 (conclusion) was revised again to make it more clear. If this section is still considered to be not acceptable, may we trouble you for some specific instructions on how Section 5 could be improved?
Once again, thank you for your patient and thoughtful review of our paper.